# Genomic variants from RNA-seq for goats resistant or susceptible to gastrointestinal nematode infection

**Hadeer M. Aboshady**[1,2,3,4], **Nathalie Mandonnet**[4], **Anna M. Johansson**[3],
**Elisabeth Jonas**[3], **Jean-Christophe Bambou**[4]*

**1** Department of Animal Production, Faculty of Agriculture, Cairo University, Cairo, Egypt, **2** AgroParisTech, Paris, France, **3** Department of Animal Breeding and Genetics, Swedish University of Agriculture Science, Uppsala, Sweden, **4** URZ, Unité de Recherches Zootechniques, INRA, Petit Bourg (Guadeloupe), France

\* jean-christophe.bambou@inrae.fr

**Data Availability Statement:** Raw sequence reads uploaded to a stable, public repository (NCBI BIOPROJECT) http://www.ncbi.nlm.nih.gov/

## Abstract

Gastrointestinal nematodes (GIN) are an important constraint in small ruminant production. Genetic selection for resistant animals is a potential sustainable control strategy. Advances in molecular genetics have led to the identification of several molecular genetic markers associated with genes affecting economic relevant traits. In this study, the variants in the genome of Creole goats resistant or susceptible to GIN were discovered from RNA-sequencing. We identified SNPs, insertions and deletions that distinguish the two genotypes, resistant and susceptible and we characterized these variants through functional analysis. The T cell receptor signalling pathway was one of the top significant pathways that distinguish the resistant from the susceptible genotype with 78% of the genes involved in this pathway showing genomic variants. These genomic variants are expected to provide useful resources especially for molecular breeding for GIN resistance in goats.

## Introduction

Gastrointestinal nematodes (GIN) are one of the most pathogenic parasites in sheep and goats that cause large economic losses. The wide geographic distribution and increasing resistance against anthelmintic molecules require alternative control strategies [1]. Selection for resistant animals using genetic information is a promising strategy. However, selection based on phenotypic traits such as faecal egg count (FEC) has been successfully used [2–4], although this strategy implements a certain degree of uncertainty since FEC is an indirect measure of resistance. The measurements of FEC is also time consuming and costly as it requires animals to be challenged with parasites either naturally or experimentally. On the other hand, selection using information from the genome could provide a faster and more sustainable tool in breeding for GIN resistance. The identification of genomic variation is a main step to understand the relationship between genotype and phenotype. Single nucleotide polymorphisms (SNPs) showed potential as molecular markers to link genotypes with desired traits in goats, such as milk

bioproject/667825 Accession: PRJNA667825 ID: 667825.

**Funding:** This study was funded by the Project MALIN (La Région Guadeloupe and Fonds Européens FEDER). H.M.A was supported by a doctoral fellowship from the project European Graduate School in Animal Breeding and Genetics.

**Competing interests:** The authors declare no competing interests.

quality and quantity [5–7], litter size [8, 9], growth rate [8, 10], fiber quality [11–13] and disease resistance [14].

Recently, Piskol et al. [15] showed that genomic variants detected from RNA sequencing (RNA-seq) data offers a cost-effective and reliable alternative for SNP discovery. They showed that among the SNPs called from the RNA-seq data, more than 98% were also identified by whole-genome sequencing or whole-exome sequencing approaches. RNA-seq technology was developed primarily for mapping and quantifying transcriptomes to analyze global gene expression in different tissues. Besides allowing the detection of differentially expressed genes, since RNA-seq functional genes are sequenced at high coverage, the search for full scale variants (SNPs, insertions and deletions) in coding genes can be performed. Up to date, this technique has been used as a method to detect SNPs in transcribed regions in an efficient and cost-effective way [13, 16–19]. In this study, we performed a genomic variant discovery analysis in the abomasal mucosa transcriptomes of resistant and susceptible Creole goats experimentally infected with *Haemonchus contortus*, chosen for its prevalence in sheep and goats throughout the temperate and tropical region of the world, and characterized the variants identified.

## Materials and methods

### Ethics statement

All animal care handling techniques and procedures as well as the procedures for experimental infection, tissue sampling and slaughtering were approved by the French Ethics Committee n˚ 069 (Comité d'Ethique en Matière d'Expérimentation Animale des Antilles et de la Guyane, CEMEAAG) authorized by the French Ministry of Higher Education, Research and Innovation. The experiment was performed at the INRA Experimental Facilities PTEA (Plateforme Tropicale d'Expérimentation sur l'Animal) according to the certificate number A 971-18-02 of authorization to experiment on living animals issued by the French Ministry of Agriculture.

### Animals and experimental design

The pedigree of each animal of the experimental flock used in this study was available from the foundation generation of 1979. In addition, since 1995, faecal samples have been regularly collected (weeks 6 and 7 after drenching) from each animal at 11 months of age for genetic evaluation on the average of 2 FEC measures. This genetic evaluation provides a breeding value (BV) for FEC of each goat of the experimental flock of PTEA, based on the FEC of their parents consolidated with their own when available (i.e. at 11 months old). The Eight 9-month old Creole kids were chosen on the basis of their extreme estimated BV (the 4 kids with the highest BV for resistance and the 4 kids with the highest BV for susceptibility) in their cohort (n = 89). Before the experiment the kids were reared at pasture, from birth to 9 months old, with a limited level of GIN contamination (FEC < 500). The FEC of the 8 kids (n = 4 resistant and n = 4 susceptible), were not statistically different at 9 months old. The average BV of the 2 groups of animals were separated by 1.04 genetic standard deviation. The animals were drenched with moxidectine (Cydectine®, Fort Dodge Veterinaria S.A., Tours, France, 300 μg/kg) and housed indoors under worm-free conditions in a single pen, one month before the start of the experiment. At 10 month old, kids were orally infected (challenge 1) with a single dose of 10,000 *H. contortus* third-stage larvae (L3) then drenched 5 weeks later with moxidectine. After 4 weeks of resting under parasite-free conditions, a fistula was surgically implanted in the abomasum of each animal to allow abomasal mucosa sampling. Four weeks after, the surgical procedure, the animals were orally infected (challenge 2) with a single dose of 10,000 *H. contortus* L3. During the challenge 2, abomasal mucosa biopsies were taken from each animal at 0, 8, 15 and 35 days post infection (dpi). Besides, faeces samples were taken twice a

week during second challenge. The details for faeces sampling, FEC measurements and data analysis have been previously published [20].

## Surgical procedure

The custom designed abomasal cannula consisted of a flexible rubber tube with a length of 7 cm and a diameter of 2 cm with a rounded base of 4 cm in diameter. This flexible plastic was chosen to limit the possibility of mechanical abrasion of the mucosal surface of the abomasum. The animals were fasted 16 h before cannula insertion surgery. The animals were premedicated with ketamine (2mg/kg IV, Le Vet Pharma, Wilgenweg, Neitherlands), xylazine (0.2mg mg/kg IM, Le Vet Pharma, Wilgenweg, Neitherlands) and oxytetracycline (20 mg/kg IM, Eurovet Animal Health, Handelsweg, Neitherlands). The animals were positioned in left lateral recumbency. Skin over the surgical site was shaved and prepped with povidone iodine (Vétédine, Laboratoire Vetoquinol S.A., Lure, France). A ventral midline incision was made to locate and exteriorize the abomasum. A 3 cm purse-string suture (Silk 2–0) was placed midway between the lesser and greater curvature and a stab incision was made in the center to insert the cannula. Then, the purse-string suture was tightened and tied off. To maintain the abomasum in an anatomically correct position, another stab incision was made in the abdominal wall at 10cm from the laparotomy incision on the right paramedian area to enable the cannula to be passed freely through. An external flange was placed over the external part of cannula and fixed with adhesive fabric plaster strip. A sterile compress was inserted into the cannula as stopper. After the surgical procedure, all the animals were housed individually with free access to fresh water and hay.

## Biopsy sampling procedure

Biopsy specimens were taken from the abomasal mucosa using a flexible endoscope (FG-24V, Pentax, France). The biopsies samples of 2×2×2mm taken with the endoscopic forceps with window (model KW1815S) were quickly snap frozen into liquid nitrogen and stored at -80˚C until RNA extraction. The animals were restrained in a harness made with a surgical drape allowing animal legs to protrude and which exposed the cannula. No sedation was used since no signs of discomfort or pain were observed during or after the procedure. The sterile compress inserted into the cannula was removed and the abomasal contents collected. The endoscope was introduced into the abomasal lumen and 3 biopsies per animal and per time points were taken from the abomasal folds of the fundic mucosa. At each time point the whole fundic mucosa was observed and no sign of mucosal injury due to the previous sampling was observed.

## RNA extraction and sequencing

Total RNA was extracted using the NucleoSpin® RNA isolation kit (Macherey-Nagel, Hoerdt, France) following the manufacturer's instructions, except that DNase digestion was performed with twice the indicated amount of enzyme. The total RNA concentration was measured with NanoDrop 2000 (ThermoScientific TM, France). The RNA integrity was verified using an Agilent Bioanalyzer 2100 (Agilent Technologies, France) with a RNA Integrity Number of > 7.5. The extracted total RNA was stored at -80˚C until sequencing.

High-quality RNA from all samples was processed for the preparation of cDNA libraries using an Illumina TruSeq RNA sample prep kit for mRNA analysis following the Illumina's protocols. After quality control and quantification, cDNA libraries were pooled in groups of 6 technical replicates and sequenced on 5 lanes on the HiSeqTM 2000 (Illumina® NEB, USA) to obtain approximatively 30 million reads (100 bp paired-end) for each sample with insert sizes ranging from 200 to 400 base pairs.

## Bioinformatics analysis

The quality control check on raw reads in FASTQ format was processed using FASTQC. The remaining reads were aligned to the *Capra hircus* genome (assembly ARS1 from NCBI) using the Burrows-Wheeler Aligner (BWA). Genomic variants including SNPs as well as small insertions and deletions (indels) were detected using mpileup in SAMtools. Variant filtering criterion: A detected variant was kept only if met four criteria: the read depth was more than 10, the quality score was over 20, the minor allele frequency was over 0.05 and the variants present at least in 50% of the group individuals and replicates to ensure the constant of each variant between individuals of the same group (group specific variant).

## Variant statistics and functional annotation

SNPs, insertions and deletions were compared between samples from the resistant and susceptible groups. Common variants were excluded and only different variants between the two groups were kept for the subsequent analysis. Genes ID and its position on chromosome (start and end position) were extracted from NCBI gff file (GCF_001704415.1_ARS1_genomic.gff) using Unix commands then merged with group discovered variants files depending on variant position to define genes containing variants. Variant information for distribution among genes was calculated within the R program after merging variants with the respective annotated genes. The effect of the variants (SNPs, insertions and deletions) on genes were determined using variant effect predictor (VEP) web interface tool provided by Ensembl online tools (https://www.ensembl.org/info/docs/tools/vep/index.html) within the goat genome reference (Assembly: ARS1) and results were extracted as .txt file for graphical interface using the R program. Genes containing variants were annotated with Gene Ontology (GO) terms under the categories of biological processes, molecular functions, and cellular components using clusterProfiler R package [21]. The Bonferroni-corrected P-value $\leq 0.05$ was used as the threshold. Additionally enriched KEGG pathways for genes containing variants were identified using the same package. The Pathview package was used for visualization [22]. The R version 3.5.1 was used.

# Results

## Phenotypic assessment of resistant and susceptible animals

Animals were tested for FEC twice a week during the challenge 2 to assess the resistant and susceptible status of the animals. Results showed that at 21 dpi there was no difference in FEC between resistant and susceptible animals. Meanwhile, FEC was significantly lower in resistant compared to susceptible animals from 23 dpi till end of the second challenge (35 dpi). This confirmed the animal status as resistant or susceptible to GIN infection with significant differences for FEC between groups.

## Sequencing and SNP discoveries

The RNA-seq produced an average of 15.3 million raw reads per sample. Our read alignment results showed that 99.3% sequencing reads (15.2 million) were successfully aligned to the ARS1 goat's reference genome with an average 79% paired-end sequencing.

Using RNA-seq reads a total of ~2.33 and ~1.82 million raw genomic variant positions expressed in the abomasal mucosa of resistant and susceptible animals were detected at different time points of infection (Table 1). After variant filtering analysis, we were able to identify 354,598 and 253,218 SNP, 20,463 and 15,645 insertion and 20,397 and 14,841 deletion records for resistant and susceptible kids, respectively. These variants were then used to produce Venn

**Table 1. Number of SNPs and indels called from transcriptome data.**

| Genotype | Days after infection | Raw | | | Filtered | | |
|---|---|---|---|---|---|---|---|
| | | SNPs | Insertion | Deletion | SNPs | Insertion | Deletion |
| Resistant | 0 | 712902 | 37798 | 43176 | 300268 | 14956 | 16841 |
| | 8 | 747086 | 38536 | 43672 | 365088 | 17179 | 20179 |
| | 15 | 818301 | 42407 | 48589 | 367130 | 17746 | 20267 |
| | 35 | 973405 | 49260 | 56391 | 416626 | 19667 | 22492 |
| | **All times** | **2107899** | **104250** | **124743** | **354598** | **20463** | **20397** |
| Susceptible | 0 | 479135 | 26102 | 29129 | 186577 | 9572 | 10689 |
| | 8 | 680515 | 35922 | 40202 | 324887 | 15288 | 17953 |
| | 15 | 753841 | 40504 | 44964 | 327288 | 16336 | 18546 |
| | 35 | 575608 | 30821 | 33198 | 245840 | 12025 | 13837 |
| | **All times** | **1635811** | **84872** | **99606** | **253218** | **15645** | **14841** |

diagrams (Fig 1) to present common and non-common variants between resistant and susceptible animals. Comparing genomic variants of the two genotypes at different time points of infection, 200,053 SNPs, 10,095 deletions and 8,755 insertions were in common (Fig 1). To explore the different genomic variants in resistant and susceptible animals, we excluded the common variants and made the subsequent analysis with non-common variants.

Non-common SNPs were annotated with the respective genes. A total of 12,142 and 8,635 genes containing SNPs were identified in the resistant and susceptible animals, respectively. In the samples from the susceptible animals an average of 5.19 SNPs were identified per gene. Meanwhile, the double number of SNPs (10.53 SNPs per gene) were identified in the data from the resistant animals. Among genes containing SNPs, genes with only 1 SNP were more common (1759 for resistant and 2629 for susceptible). Genes with 10 SNPs or less accounted for 69.1% and 88.9% of all SNPs identified in the samples from the resistant and susceptible animals, respectively. Results for SNPs distributions among genes showed that more genes containing one or two SNPs were identified in the susceptible animals, while more genes containing 3 or more SNPs were identified in the resistant animals (Fig 2).

Following the same procedure, non-common insertions and deletions were annotated to genes. A total of 4848 and 3292 genes containing insertions and 4660 and 2610 genes containing deletions were identified in data from the resistant and susceptible animals, respectively. Among them, genes with one insertion or deletion were more common and accounted for more than 50% of all insertions or deletions identified. Insertion and deletion distributions

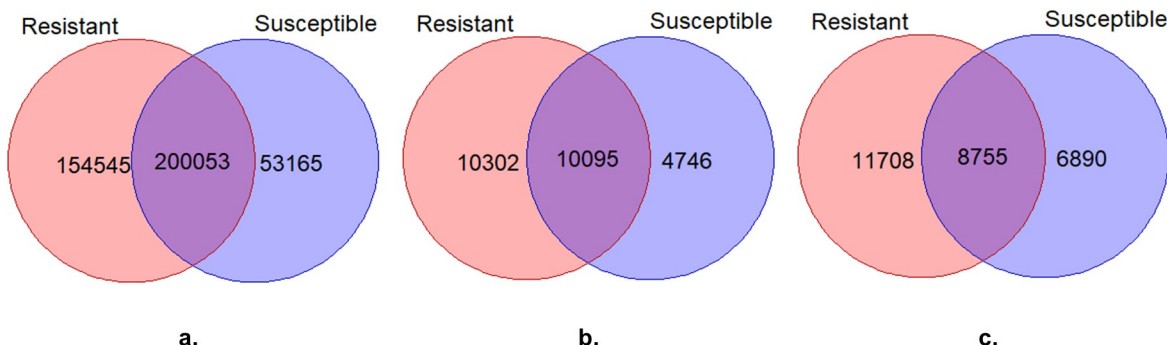

**Fig 1.** Venn diagram for SNP (a) deletion (b) and insertion (c) variants identified in samples from the resistant and susceptible group.

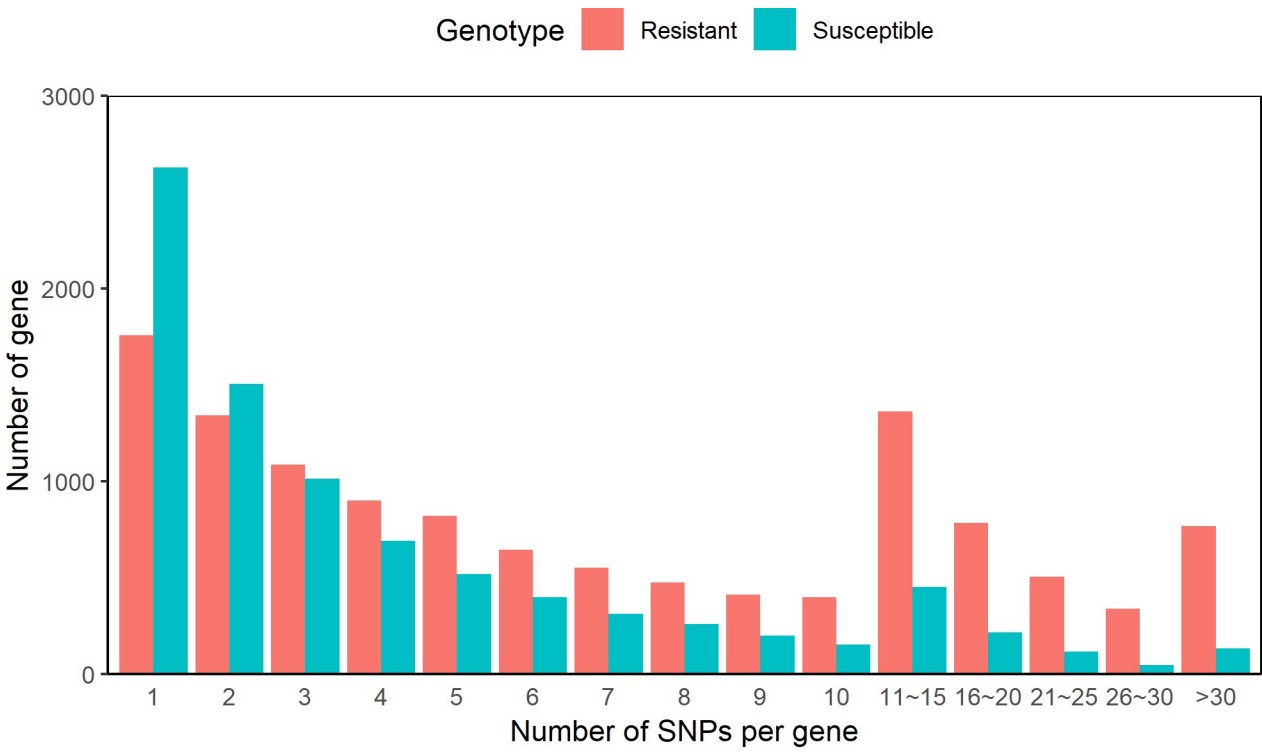

**Fig 2. Single nucleotide polymorphism (SNP) distribution among genes.**

among genes are shown in Fig 3. Data from the resistant animals always contained more insertions and deletions among genes.

All variants (SNPs, insertions and deletions) were combined in two files, one for variants identified in data from resistant animals and one for variants identified in data from susceptible animals. A variant effect prediction analysis was made for both files and the results are shown in Fig 4. The highest variants ratio was around 50% for intron variants from variants in the resistant (56%) and susceptible (47%) animals, followed by downstream and upstream gene variants (20% and 8% in resistant and 25% and 8% in susceptible animals). A total of 3% of missense variants were predicted in the resistant and 4% in the susceptible animals (Fig 4).

After the GO enriched analysis, 10,736 and 8,538 genes containing genomic variants in the resistant and susceptible animals were assigned with one or more GO terms. The top 10 significant GO terms under the categories of biological processes, molecular functions, and cellular components of these annotated genes are shown in Fig 5. Six to seven of the top 10 terms under each GO category were similar whatever the genotype. For the cellular component, the major category that was identified in the data from the resistant but not the susceptible animals was the mitochondrial matrix. For the molecular function, phosphoric ester hydrolase activity was the most represented term for the variants identified in data from the resistant animals which were not present in data from the susceptible ones. The most significant GO terms in the category of biological process, were phospholipid metabolic process and macroautophagy which were identified only in the data from the resistant animals.

The top 10 significant KEGG pathways for genes containing genomic variants identified in data from both genotypes are presented in Fig 6. Mitogen-activated protein kinase (MAPK) signaling pathway, T cell receptor (TCR) signaling pathway, hepatitis B and longevity regulating pathway were the top significant pathways identified only in data from the resistant

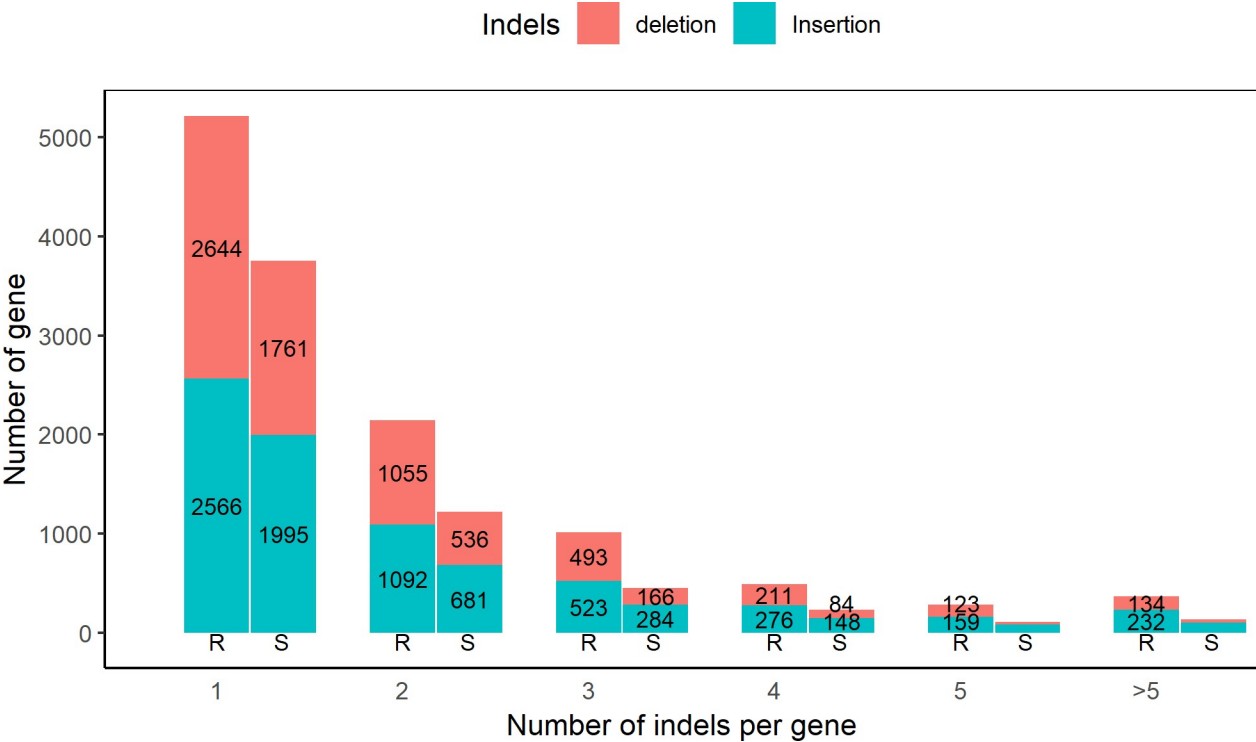

**Fig 3. Insertions and deletions (indels) identified among genes using the data from resistant (R) and susceptible (S) kids.**

animals. Activation of T lymphocytes is a key event of the host adaptative immune response. Therefore, we focused on TCR signaling pathway that was identified in the top KEGG

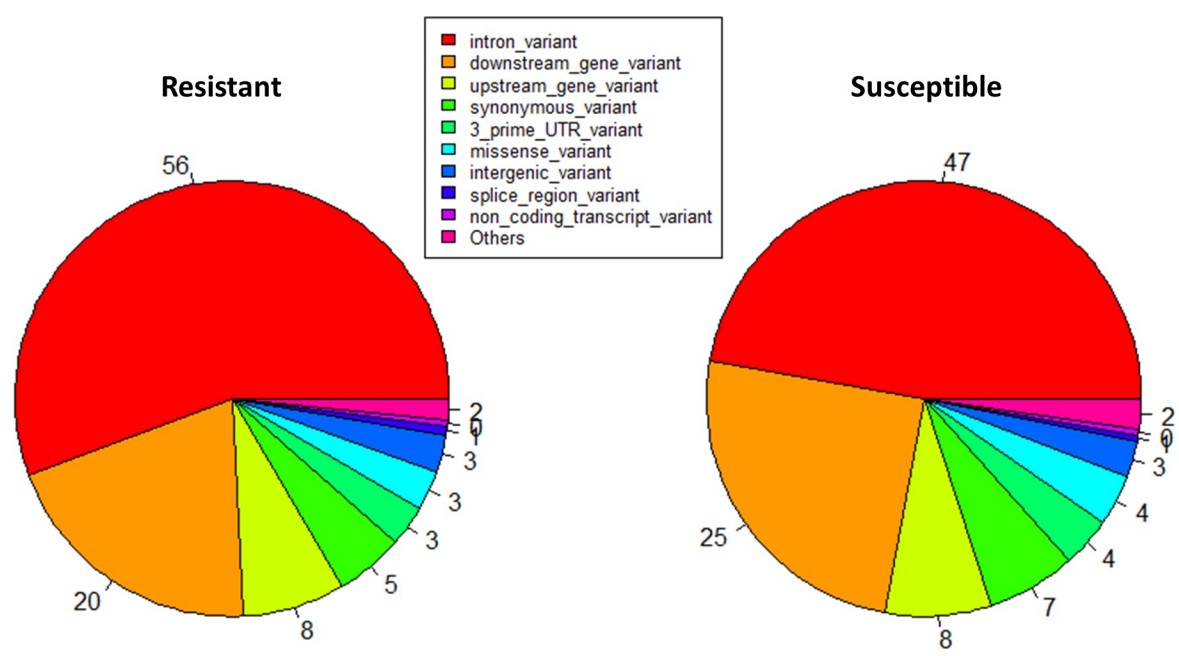

**Fig 4. Variant effect prediction in resistant and susceptible animals.**

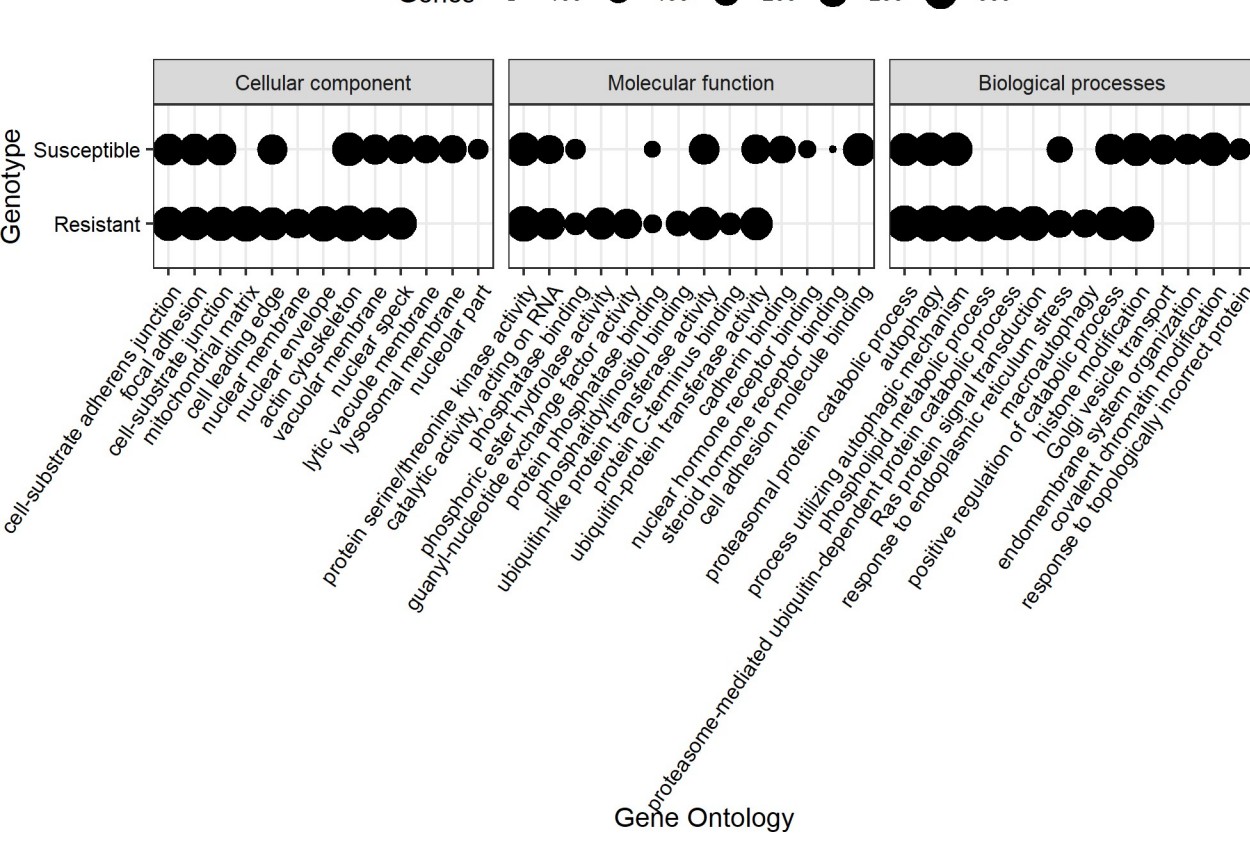

**Fig 5. Gene ontology for genes containing variants for data from the resistant and susceptible genotypes.**

pathways for the resistant animals. The TCR signaling pathway and the number of genomic variants for each gene in this pathway in the data from the resistant animals are presented in Fig 7. A total of 100 genes are known to control this pathway in *C. hircus* breeds. A least one genomic variants was identified in 78 genes out of these 100 genes (Fig 7).

## Discussion

RNA-seq technology provides a deep and efficient way to study whole transcriptome which involve the complete set of transcripts at a corresponding developmental stage or physiological condition. RNA-seq has been used widely in quantifying transcriptomes to analyze global gene expression in different tissues [23–26]. Besides that, recently it has been shown that RNA-seq provides a resource for gene-associated SNP discovery [13, 19, 27]. In this study we used for the first time RNA-seq for the discovery of SNP associated with resistance to GIN in goats. One major problem for goats and sheep genomic analysis is that the functional analysis was not available with goats or sheep as reference species due to the lack of data for *C. hircus* and *Ovis aries* in functional analysis programs. Therefore, almost all previous publications in goats and sheep genomic used human [28–30] or bovine [31] genome as reference for functional analysis. Nowadays, the availability of the goat genome reference within Ensembl and KEGG pathways made the variants effect prediction and pathways analysis possible and more specific for goats. Our study is one of the first studies, to use the goat genome as reference in the functional analysis. Besides, we used a goat breed from a tropical region, where 80% of the world

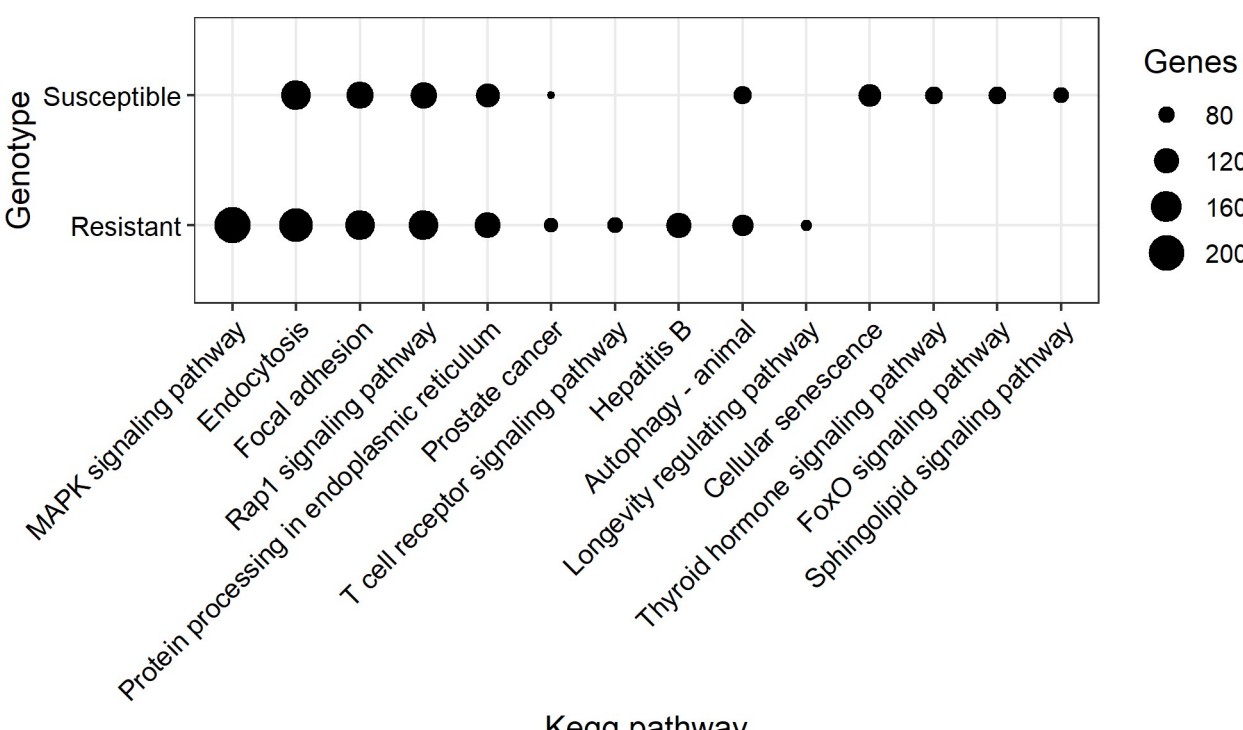

**Fig 6. KEGG pathways for genes containing genomic variants in the data from the resistant and susceptible animals.**

goat population is found and provides one main source of animal protein [32] (http://www.fao.org/faostat/).

Association analysis studies are usually performed to identify genomic variants that affect phenotypic variation for resistance to GIN. Genome-wide association study in goats, identified seven SNPs associated with GIN resistance and the genes near to these positions were related to the immune response and other functions [33]. Meanwhile, other studies performed candidate gene SNP analysis, mainly immune genes, to identify SNPs associated with resistance to GIN in goats [34, 35]. A limit for SNP association studies for parasite resistance is that it requires genotyping and phenotyping of large numbers of animals [36]. Recently, gene expression profiling via RNA-seq revealed hundreds of genes that involve in immune function for goats resistance to GIN [20, 29, 37]. Through RNA-seq, besides allowing gene expression profiling, functional genes are sequenced at high coverage, allowing to full scale variants discovery in coding genes. This technique has been used as a methods to detect SNPs in transcribed regions in an efficient and cost-effective way in goats for climate adaptation traits [17] and fiber quality [13]. Climate adaptation traits were studied by searching for gene-associated SNPs from RNA-seq of liver and kidney samples of Indian goat breeds from different climatic region and the different biological processes and gene networks involved were analysed [17]. In addition, a large number of putative SNPs were identified by the analysis of RNA-seq of the Cashmere goat's skin and functional annotation for these SNPs were carried out to indicate different pathways involved in fiber development and quality [13]. In the present study, we used data from RNA-seq of abomasal mucosa samples from GIN resistant and susceptible Creole goats at four time point after *H. contortus* infection to identify putative gene-related genomic variants. To our knowledge, this is the first study to identify genomic variants from RNA-seq data in goats infected with a GIN.

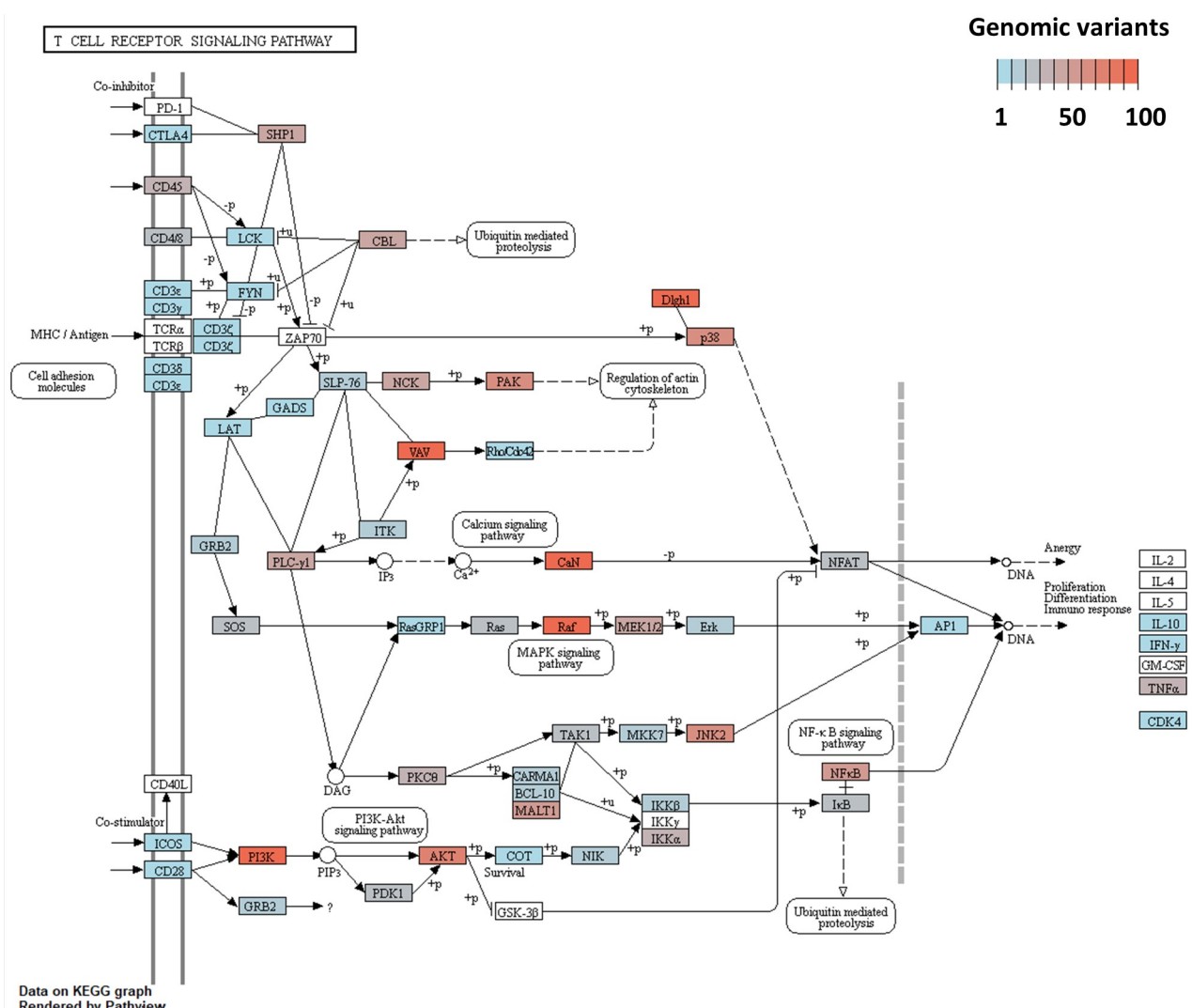

**Fig 7. Number of genomic variants for each gene in the T cell receptor signalling pathway in the resistant animals.**

One benefit to variant calling from RNA-seq is the focus on genes/transcripts that are expressed in the target tissue. However, there are some concerns about the information provided by such data. It was previously shown that some false positive SNPs identified in cDNA arise from alignment of a read to the wrong gene which represents a problem in gene families with highly conserved domains when using short sequence reads [16, 38]. This situation has also been observed in regions associated with sequence repeats [39]. Therefore, one great challenge of using Illumina sequencing for transcriptome analysis is the short read length. We have used the Illumina TruSeq RNA sample prep kit that generated read lengths of $2 \times 75$ bases pairs with paired-end reads to increase the base coverage within expressed genes in a sample and as a result improved variant detection sensitivity.

Our study revealed many variants in intronic regions. Indeed, it has been shown that the percentage of introns reads was higher when libraries were prepared with total RNA extraction methods, as in the present study, compared with the poly(A) RNA ones [40]. It should also be noted that the results presented here for variants position do not necessarily reflect the

percentages of intron/exon, since variant effect prediction had not been made for all variants but only for those which differ between resistant and susceptible animals.

Studies for GIN infection in cattle indicated that GO terms associated with genes that were differently expressed between resistant and susceptible cattle were predominantly related to lipid metabolism and the top function of regulatory networks identified was associated with lipid metabolism [41]. Our results showed that the most significant GO term associated with genes containing variants in resistant animals was phospholipid metabolic process. T cells are a subset of lymphocytes that have a central role in adaptive immune response. The TCR is a complex of integral membrane proteins on the surface of T cells, which takes part in the activation of T cells in response to antigen recognition and eventually results in cellular proliferation, differentiation, cytokine production, and/or activation-induced cell death [42]. In the present study, we found that genes containing genomic variants in resistant goats were associated with the TCR signalling pathway in KEGG pathway enriched analysis. Interestingly, a rapid increase in γδ-TCR+ T cells numbers within the abomasal mucosa at 3 and 5 days after *H. contortus* infection has previously been shown in sheep [43, 44]. These results suggest that this early TCR signalling coupled with the formation of a stable and specialized junction between T cell and antigen-presenting cells would be involved in the protective immune response against GIN infection [42].

Mitogen-activated protein kinases (MAPK) are highly conserved key regulators of cellular transduction systems, activated notably by growth factors, hormones and cytokines to mediate cell growth, apoptosis, stress response and immune response in mammalian cells [45]. The MAPK signalling pathway was in the top significant pathways identified from genes containing genomic variants of the resistant genotype. The activation of this pathway was associated with differently expressed genes identified in the blood transcriptome and the abomasal mucosa respectively in goats and sheep, suggesting a key role of the MAPK signalling in the mechanisms of resistance to GIN [29, 46].

## Conclusion

This study suggests that the TCR signalling pathway plays an important role in GIN resistance in goats and provides valuable resources for the characterization of genomic markers for GIN resistance in goats. Furthermore, here we showed the possibility to use RNA-seq data as an efficient and cost-effective method to detect genomic variants in transcribed genomic regions. This additional use of such high throughput data could constitute a great resource to gain further knowledge of animal resources.

## Acknowledgments

The authors want to give thanks to the Duclos team for care and handling of the animals: F. Pommier, F. Nimirf, F. Periacarpin, C. Deloumeaux and M. Jean-Bart. A special thanks to Dr. Harry Archimede for his valuable contribution for the surgical procedure.

## Author Contributions

**Conceptualization:** Hadeer M. Aboshady, Nathalie Mandonnet, Jean-Christophe Bambou.

**Formal analysis:** Hadeer M. Aboshady, Jean-Christophe Bambou.

**Funding acquisition:** Nathalie Mandonnet, Elisabeth Jonas, Jean-Christophe Bambou.

**Investigation:** Jean-Christophe Bambou.

**Supervision:** Nathalie Mandonnet, Anna M. Johansson, Elisabeth Jonas, Jean-Christophe Bambou.

**Writing – original draft:** Hadeer M. Aboshady.

**Writing – review & editing:** Anna M. Johansson, Elisabeth Jonas, Jean-Christophe Bambou.

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
