## [Decision Letter · Decision Letter 0]

28 Sep 2020

PONE-D-20-03648

Genomic variants from RNA-seq for goats resistant or susceptible to gastrointestinal nematode infection

PLOS ONE

Dear Dr. Bambou,

Thank you for submitting your manuscript to PLOS ONE. After careful consideration, we feel that it has merit but does not fully meet PLOS ONE’s publication criteria as it currently stands. Therefore, we invite you to submit a revised version of the manuscript that addresses the points raised during the review process.

We look forward to receiving your revised manuscript.

Kind regards,

Raluca Mateescu

Academic Editor

PLOS ONE

Journal Requirements:

2. We note that you are reporting an analysis of a microarray, next-generation sequencing, or deep sequencing data set. PLOS requires that authors comply with field-specific standards for preparation, recording, and deposition of data in repositories appropriate to their field. Please upload these data to a stable, public repository (such as ArrayExpress, Gene Expression Omnibus (GEO), DNA Data Bank of Japan (DDBJ), NCBI GenBank, NCBI Sequence Read Archive, or EMBL Nucleotide Sequence Database (ENA)). In your revised cover letter, please provide the relevant accession numbers that may be used to access these data. For a full list of recommended repositories, see http://journals.plos.org/plosone/s/data-availability#loc-omics or http://journals.plos.org/plosone/s/data-availability#loc-sequencing.

Reviewers' comments:

Reviewer's Responses to Questions

**Comments to the Author**

1. Is the manuscript technically sound, and do the data support the conclusions?

Reviewer #1: Yes

Reviewer #2: Partly

2. Has the statistical analysis been performed appropriately and rigorously? 

Reviewer #1: Yes

Reviewer #2: N/A

3. Have the authors made all data underlying the findings in their manuscript fully available?

Reviewer #1: No

Reviewer #2: Yes

4. Is the manuscript presented in an intelligible fashion and written in standard English?

Reviewer #1: Yes

Reviewer #2: No

5. Review Comments to the Author

Reviewer #1: This is a clear and interesting manuscript by a highly regarded group. However, some aspects of the methodology are not clear.

On line 71 the size of the cohort should be given.

Line 121 Groups of six what? Are they replicates from he same animal or were animals pooled?

Lines 131-132 The justification for a frequency of over 50% is not clear.

Reviewer #2: Article entitled, “Genomic variants from RNA-seq for goats resistant or susceptible to GI nematode infection is not acceptable in the present format and demands thorough revision. Although the content of the article is good enough to publish in PLOS ONE but the MS has been written in a very casual way. I request the authors to take up the challenge to revise the MS thoroughly for improvement. The following points should be taken into consideration:

1. Assignment of resistant and susceptible animals is confusing. The authors are requested to furnish details on this point.

2. In Discussion the meaning of first sentence is not clear. It appears that, they have worked with resistant/susceptible nematodes.

3. In my opinion total reorientation of Discussion is warranted.

(a) First paragraph should highlight the novelty of work stating research gap. Subsequently the authors should speak about the importance of the study and its uniqueness compared to other techniques for identification of resistant and susceptible animals.

(b) Second paragraph should emphasize “how SNPs have been exploited in caprine for climate adaptations study and other economic trait. Subsequently the authors are requested to mention other genetic studies to identify GIN resistant breeds. To mention a few are: Llie et al., 2018 (study in Romania and Hungary), Alam et al., 2019 published in J Anim Sci Biotechnol and Sharma et al., 2012 published in International J. Anim. Vet. Sci. Apart from these if the authors find few more publications are requested to add in the Discussion.

(c) Third paragraph should emphasize the role of T cells for conferring adaptive immunity against GIN.

(d) Existing 3rd paragraph sounds good and I feel this should be retained. However, I will be curious to see an elaboration of MAPK signaling pathway with a brief note on the observation in Yichang white goats.

Overall Assessment: Worthy to publish in PLOS ONE if the corrections are incorporated.

6. PLOS authors have the option to publish the peer review history of their article (what does this mean?). If published, this will include your full peer review and any attached files.

Reviewer #1: No

Reviewer #2: No

---

## [Author Response · Author response to Decision Letter 0]

29 Oct 2020

Revision Notes

Manuscript # PONE-D-20-03648

Dear Editor,

 First, we would like to thank you for giving us the chance for revision and to respond to the reviewers' comments. We have modified the manuscript according to the reviewers' recommendations and answered their questions in detail.

Reviewer #1

1. Reviewer/ This is a clear and interesting manuscript by a highly regarded group. However, some aspects of the methodology are not clear.

R/ we made some changes to the methodology to make it more clear according to the raised points from the reviewer 1 and reviewer 2 as mentioned in details subsequently.

2. Reviewer/ On line 71 the size of the cohort should be given.

R/ the size of the cohort have been added (n=89) # line 74

3. Reviewer/ Line 121 Groups of six what? Are they replicates from the same animal or were animals pooled?

R/ The sentence has been corrected. In the revised version # lines 126-127 “groups of 6 technical replicates”

4. Reviewer/ Lines 131-132 The justification for a frequency of over 50% is not clear.

R/ we added justification. In the revised version # lines 137-138. “to ensure the constant of each variant between individuals of the same group (group specific variant)”

Reviewer #2 

1. Reviewer/ Article entitled, “Genomic variants from RNA-seq for goats resistant or susceptible to GI nematode infection is not acceptable in the present format and demands thorough revision. Although the content of the article is good enough to publish in PLOS ONE but the MS has been written in a very casual way. I request the authors to take up the challenge to revise the MS thoroughly for improvement. The following points should be taken into consideration.

R/ we improved the revised manuscript taking into consideration the raised points from the reviewers.

2. Reviewer/ Assignment of resistant and susceptible animals is confusing. The authors are requested to furnish details on this point.

R/ we added details on this point. # lines 67-74 and lines 155- 159

3. Reviewer/ In Discussion the meaning of first sentence is not clear. It appears that, they have worked with resistant/susceptible nematodes.

R/ we modified the sentence. # lines 273- 275. 

4. Reviewer/ In my opinion total reorientation of Discussion is warranted.

(a) First paragraph should highlight the novelty of work stating research gap. Subsequently the authors should speak about the importance of the study and its uniqueness compared to other techniques for identification of resistant and susceptible animals.

R/ The raised point have been taken into account # lines 240 -255

(b) Second paragraph should emphasize “how SNPs have been exploited in caprine for climate adaptations study and other economic trait. Subsequently the authors are requested to mention other genetic studies to identify GIN resistant breeds. To mention a few are: Llie et al., 2018 (study in Romania and Hungary), Alam et al., 2019 published in J Anim Sci Biotechnol and Sharma et al., 2012 published in International J. Anim. Vet. Sci. Apart from these if the authors find few more publications are requested to add in the Discussion.

R/ second paragraph have been rebuilt taking into account the raised point. # lines 256 -277

(c) Third paragraph should emphasize the role of T cells for conferring adaptive immunity against GIN.

R/ The raised point have been taken into account # lines 288 -302

(d) Existing 3rd paragraph sounds good and I feel this should be retained. However, I will be curious to see an elaboration of MAPK signaling pathway with a brief note on the observation in Yichang white goats. 

R/ The raised point have been taken into account # lines 303 -310

Journal Requirements:

Raw sequence reads uploaded to a stable, public repository (NCBI BIOPROJECT)

http://www.ncbi.nlm.nih.gov/bioproject/667825

Accession: PRJNA667825 

ID: 667825

---

## [Decision Letter · Decision Letter 1]

1 Feb 2021

PONE-D-20-03648R1

Genomic variants from RNA-seq for goats resistant or susceptible to gastrointestinal nematode infection

PLOS ONE

Dear Dr. Bambou,

Thank you for submitting your manuscript to PLOS ONE. After careful consideration, we feel that it has merit but does not fully meet PLOS ONE’s publication criteria as it currently stands. Therefore, we invite you to submit a revised version of the manuscript that addresses the points raised during the review process.

We look forward to receiving your revised manuscript.

Kind regards,

Raluca Mateescu

Academic Editor

PLOS ONE

Reviewers' comments:

Reviewer's Responses to Questions

**Comments to the Author**

1. If the authors have adequately addressed your comments raised in a previous round of review and you feel that this manuscript is now acceptable for publication, you may indicate that here to bypass the “Comments to the Author” section, enter your conflict of interest statement in the “Confidential to Editor” section, and submit your "Accept" recommendation.

Reviewer #1: All comments have been addressed

2. Is the manuscript technically sound, and do the data support the conclusions?

Reviewer #1: Yes

3. Has the statistical analysis been performed appropriately and rigorously? 

Reviewer #1: Yes

4. Have the authors made all data underlying the findings in their manuscript fully available?

Reviewer #1: Yes

5. Is the manuscript presented in an intelligible fashion and written in standard English?

Reviewer #1: No

6. Review Comments to the Author

Reviewer #1: This is a useful manuscript. I am surprised that there are so many variants within each gene and a definition of how genes were defined would be helpful. Many of the variants occur in the introns and I am surprised that the transcriptome reveals so many introns; perhaps an explanation here would be helpful?

There are some minor errors in the English:

Line 17 drop 'grazing'

Line 60 Ethics

Line 77 average

Line 78 'distant' to 'separated'

Line 89 specify type of plastic

Line 157 'The details .... ' to materials and methods

Line 159 more information required

Line 168 resistant

Lines 177 and 221 variants

7. PLOS authors have the option to publish the peer review history of their article (what does this mean?). If published, this will include your full peer review and any attached files.

Reviewer #1: No

---

## [Author Response · Author response to Decision Letter 1]

23 Feb 2021

Reviewer #1

Reviewer/ This is a useful manuscript. I am surprised that there are so many variants within each gene and a definition of how genes were defined would be helpful. Many of the variants occur in the introns and I am surprised that the transcriptome reveals so many introns; perhaps an explanation here would be helpful?

R/ A definition of how genes were defined have been added. In the tracked changes version # lines 144-147

An explanation for the result that the transcriptome reveals so many introns have been added. In the tracked changes version # lines 293-298

Reviewer/ There are some minor errors in the English:

Reviewer/ Line 17 drop 'grazing'

R/ Corrected. In the tracked changes version # line 17

Reviewer/ Line 60 Ethics

R/ Corrected. In the tracked changes version # line 60

Reviewer/ Line 77 average

R/ Corrected. In the tracked changes version # line 77

Reviewer/ Line 78 'distant' to 'separated'

R/ Corrected. In the tracked changes version # line 78

Reviewer/ Line 89 specify type of plastic

R/ The cannula used was made of rubber. In the tracked changes version # line 91

Reviewer/ Line 157 'The details .... ' to materials and methods

R/ It has been transferred. In the tracked changes version # line 87-89

Reviewer/ Line 159 more information required

R/ More information have been added. In the tracked changes version # line 163-165

Reviewer/ Line 168 resistant

R/ Corrected. In the tracked changes version # line 176

Reviewer/ Lines 177 and 221 variants

R/ Corrected. In the tracked changes version # line 185 and 229

---

## [Editor Report · Decision Letter 2]

26 Feb 2021

Genomic variants from RNA-seq for goats resistant or susceptible to gastrointestinal nematode infection

PONE-D-20-03648R2

Dear Dr. Bambou,

We’re pleased to inform you that your manuscript has been judged scientifically suitable for publication and will be formally accepted for publication once it meets all outstanding technical requirements.

Kind regards,

Raluca Mateescu

Academic Editor

PLOS ONE
---

## [Editor Report · Acceptance letter]

4 Mar 2021

PONE-D-20-03648R2 

Genomic variants from RNA-seq for goats resistant or susceptible to gastrointestinal nematode infection 

Dear Dr. Bambou:

I'm pleased to inform you that your manuscript has been deemed suitable for publication in PLOS ONE. Congratulations! Your manuscript is now with our production department. 

Kind regards, 

on behalf of

Dr. Raluca Mateescu 

Academic Editor

PLOS ONE